# Machine Learning Methods for Seismic Hazards Forecast

**Valeri G. Gitis**[ID] **and Alexander B. Derendyaev** *[ID]

The Institute for Information Transmission Problems, Moscow 127051, Russia
* Correspondence: wintsa@gmail.com

**Abstract:** In this paper, we suggest two machine learning methods for seismic hazard forecast. The first method is used for spatial forecasting of maximum possible earthquake magnitudes ($M_{max}$), whereas the second is used for spatio-temporal forecasting of strong earthquakes. The first method, the method of approximation of interval expert estimates, is based on a regression approach in which values of $M_{max}$ at the points of the training sample are estimated by experts. The method allows one to formalize the knowledge of experts, to find the dependence of $M_{max}$ on the properties of the geological environment, and to construct a map of the spatial forecast. The second method, the method of minimum area of alarm, uses retrospective data to identify the alarm area in which the epicenters of strong (target) earthquakes are expected at a certain time interval. This method is the basis of an automatic web-based platform that systematically forecasts target earthquakes. The results of testing the approach to earthquake prediction in the Mediterranean and Californian regions are presented. For the tests, well known parameters of earthquake catalogs were used. The method showed a satisfactory forecast quality.

**Keywords:** machine learning; expert estimate; maximum possible magnitudes of earthquakes; one class classification; seismic hazard; seismic zoning; earthquake forecasting

---

## 1. Introduction

Tectonic earthquakes are invariably preceded by a period when stresses increase in the Earth. This process forms anomalous changes in the geological environment near the source of the expected earthquake [1–3]. To describe the seismotectonic properties of the geological environment, various types of data are used: Earthquake catalogs, time series of geodetic [4], geophysical [5] and geochemical measurements [6], and aerospace observations [7]. The success of seismic hazard forecast is largely influenced by both completeness of the description of the spatial and spatio-temporal properties of the seismic process, and the possibility of their joint analysis. In our approach to joint analysis, all available data on the properties of the process are converted into grid fields [8].

Seismic zoning is prerequisite for seismic hazard assessment [9]. The most important and complex problem of seismic zoning is to map the maximum possible magnitudes of earthquakes ($M_{max}$). The values of $M_{max}$ cannot be measured instrumentally. Two assumptions are used to construct a digital map of $M_{max}$: (1) The assumption of large earthquake repetition [10] and (2) the assumption that the values of $M_{max}$ depend on the properties of the geological environment [11,12].

The statistical approach uses only the first assumption. This means that the $M_{max}$ map is calculated using only those earthquakes whose epicenters fall into a sliding spatial window. The methods of extreme statistics are used for the estimation of $M_{max}$ [13–16]. These methods require a sufficiently large number of observations, which may be unavailable for some zones in the region. To improve this method, [17] used the second assumption, and estimated $M_{max}$ by earthquake epicenters within geologically homogeneous zones identified by an expert geologist. As in the previous approach,

this method does not provide for extrapolating $M_{max}$ values from zones in which there are many epicenters of strong earthquakes and therefore estimates of $M_{max}$ are fairly accurate, to zones with similar seismotectonic properties, but with a small number of strong earthquakes.

The history of seismic observations is very short in relation to the speed of tectonic processes, and earthquakes with magnitudes close to maximum occur relatively rarely. To compensate for this effect, attempts are made to extrapolate reliable estimates to areas with similar seismotectonic properties of the geological environment. Reference [18] used the $M_{max}$ mapping method based on a solution of a group of experts. One of the possible algorithmic approaches applies cluster analysis [19]. A cluster analysis program divides the region into zones with similar values of geological and geophysical characteristics. These zones may consist of several isolated areas. Next, the maximum magnitudes of earthquakes recorded in one area of the zone are extrapolated to all other areas. The disadvantages of this approach are related to the fact that the zoning of a region into quasi-homogeneous zones is largely determined a set of selected features, the method of measuring the similarity between clusters, the type of clustering algorithm and, finally, the criterion of stopping the clustering process.

We describe the method of approximation of interval expert estimates which is a regression approach to the construction of a forecast map of $M_{max}$ [20–22]. To compile the map, dependence of $M_{max}$ on properties of geological environments $\mathbf{x} = (x_1, \ldots, x_I)$ is used. The values at the training sample points are determined using expert knowledge. To this end, experts choose the most studied points of the region with different seismicity and geological conditions. The expert indicates the boundaries of the interval in which, in his opinion, lies the value of $M_{max}$, and evaluates the values of the confidence that $M_{max}$ cannot exceed the lower and upper limits of the interval. In the assessment, the expert uses historical seismic data, instrumental data on the maximum magnitude of an earthquake in the vicinity of the point in question and data on the properties of the geological zone to which this point belongs. The algorithm generalizes the least squares approximation algorithm.

The task of predicting an earthquake is to determine the time, location, and magnitude of a future earthquake. Earthquake prediction studies are conducted in many directions. They include the study of the rock failure and earthquake precursor phenomena, the study of stochastic models for earthquake prediction, machine learning methods, and testing earthquake prediction algorithms [1–3,23–29]. At the same time, there are a number of works in which it is stated that earthquakes cannot be predicted [30].

Here we suggest a new method of machine learning, called the method of the minimum area of alarm, and describe a web-based platform that predicts earthquakes in automatic mode (http://distcomp.ru/geo/prognosis/). Our method solves the one-class classification problem (other methods can be found, for instance, in [31–33]). Our training sample set includes rare anomalous objects (the epicenters of target earthquakes) and grid fields of properties of the seismic process (field of features). The method allows one to detect the largest number of the target earthquakes for the training set, provided that the size of the spatio-temporal alarm area does not exceed a specified value. We present the results of testing the approach on the data of the Mediterranean and California regions.

## 2. The Method of Approximation of Interval Expert Estimates

Let the seismotectonic properties of the region under study be represented by a set of spatial grid fields of features $\mathbf{X}_1, \mathbf{X}_2, ..., \mathbf{X}_I$, and the values of the maximum possible magnitudes of earthquakes ($M_{max}$) be represented by a sample set of expert estimates. The task is to find from these data the function $F(\mathbf{x})$, which approximates the values of $M_{max}$ at the sample set, where $\mathbf{x} = (x_1, ..., x_I)$ is the vector with the values of the fields of features. The $M_{max}$ map is the $F(\mathbf{x})$ values calculated for all grid nodes of a region.

The type of expert evaluation should be convenient and straightforward for unambiguous understanding by all participants of the expert survey and should enable the expert to formalize

his knowledge about the value of the forecast fully. These requirements correspond to interval expert estimates:

$$Q_{qn} = (m_{qn}^{(1)}, m_{qn}^{(2)}, w_{qn}^{(1)}, w_{qn}^{(2)}),$$ (1)

where $m_{qn}^{(1)}, m_{qn}^{(2)}$ are the interval boundaries within which all the values of $M_{max}$ at the point $n$ are the most probable and equally possible, in the opinion of the $q$-th expert, $m_{qn}^{(1)} \le m_{qn}^{(2)}$; $w_{qn}^{(1)} > 0, w_{qn}^{(2)} > 0$ are the weighs on which the $q$-th expert indicates the degree of his confidence in the possibility that the value of $M_{max}$ may be less or greater than the corresponding interval boundary $m_{qn}^{(1)}$ or $m_{qn}^{(2)}$.

We can assume that the expert estimate $Q$ corresponds to some function of the subjective probability density $f(Y, Q)$, which reflects the expert's opinion about the value of $Y$ at a given sample point. This function takes a constant value within the interval $[m^{(1)}, m^{(2)}]$ and decreases with the weights $w^{(1)}$ and $w^{(2)}$ respectively to the left and right of the interval boundaries:

$$f(Y, Q) = C \cdot \exp\{-(w^{(1)} \frac{|m^{(1)} - Y| + m^{(1)} - Y}{2} + w^{(2)} \frac{|m^{(2)} - Y| - m^{(2)} + Y}{2})^p\},$$ (2)

where $p \ge 1$, and $C$ is defined by the condition $\int\limits_{-\infty}^{\infty} f(Y, Q) dy = 1$.

Suppose that there is a training sample $\{Q_{qn}, \mathbf{x}_n\}$, where $q$ and $n$ represent the expert and sample number. It is required to approximate the function $Y(x)$ in a certain class of functions $F(\mathbf{x}, \boldsymbol{\theta}) : \boldsymbol{\theta} \in \boldsymbol{\Theta}$, where $\boldsymbol{\Theta}$ is the domain of admissible values of the vector $\boldsymbol{\theta}$.

Let's replace $Y$ in (2) with the value of the forecast function $F(\mathbf{x}, \boldsymbol{\theta})$ and consider the function

$$r(\mathbf{x}, \boldsymbol{\theta}) = -\ln f(Y, Q) + \ln C = (w^{(1)} \frac{|m^{(1)} - Y| + m^{(1)} - Y}{2} + w^{(2)} \frac{|m^{(2)} - Y| - m^{(2)} + Y}{2})^p.$$ (3)

The function $r(\mathbf{x}, \boldsymbol{\theta})$ determines the penalty for the inaccuracy of the approximation of the expert judgment $Q$ by the value $F$ of the forecast function. To estimate $\boldsymbol{\theta}$, the average penalty on the set is minimized. The estimation has the form

$$\hat{\boldsymbol{\theta}} = \arg \min_{\boldsymbol{\theta} \in \boldsymbol{\Theta}} \sum_n \sum_q r(F(\mathbf{x}_n, \boldsymbol{\theta}), Q_{qn}).$$ (4)

It is obvious that if the forecast function $F(\mathbf{x}, \boldsymbol{\theta})$ is linear in the parameters, then the functional (4) is convex. If the domain $\boldsymbol{\Theta}$ of admissible values of the vector is also convex, then it is possible to use iterative gradient algorithms for estimation.

It is easy to see from (3) and (4) that in case of $m^{(1)} = m^{(2)}$ and $w^{(1)} = w^{(2)}$ for all expert estimates the estimation algorithm (4) coincides with the method of the least absolute errors for $p = 1$, and with the method of least squares for $p = 2$. It was shown in [22] that under certain assumptions, the estimate (4) is an estimate of the maximum likelihood.

The method of approximation of interval expert estimates was repeatedly used to construct the maps of $M_{max}$ in a number of regions, in particular, Bulgaria [34], Caribbean and Middle America Region [35], Central Europe [12,36], Costa Rica [37], the Caucasus [21], and North Caucasus [38]. In these papers, the dependences $M_{max}(\mathbf{x})$ were always estimated in a class of the sum of piecewise linear functions of geological and geophysical features. This estimation allows one to interpret the $M_{max}$ map as the sum of non-linearly transformed fields of features.

For each of the above regions, from 10 to 200 geological and geophysical fields were analyzed. 3–4 of the most informative fields were selected from this set using the stepwise regression method. Prediction functions are the sum of piecewise linear dependencies on the values of these fields. The sum of nonlinearly transformed fields defines the $M_{max}$ field. This is convenient for the seismotectonic interpretation of the $M_{max}$ map.

Interpretation of the $M_{max}$ map by a specialist allows a qualitative assessment of the accuracy of determining the maximum possible seismic hazard but is not an assessment of its accuracy. By definition, $M_{max}$ values cannot be measured instrumentally. Statistical estimates of $M_{max}$ estimates are possible only in areas with sufficiently high seismic activity. In the considered method, the values of $M_{max}$ are replaced by interval expert estimates. These estimates are approximated by a nonlinear function of geological and geophysical fields. The accuracy of the $M_{max}$ forecast is determined by the deviations of the $M_{max}$ forecast map values from expert estimates. For the above regions, from 100 to 400 expert evaluations were used. The number of parameters estimated during training ranged from eight to 14 in each region. For those regions, who did not participate in the training, the average approximation errors of expert estimates are in the range from 0.2 to 0.34.

### 3. Method of the Minimum Area of Alarm

Let the properties of the seismic process are described by the spatial and spatio-temporal fields of features in a single coordinate grid with a step $\Delta x \times \Delta y \times \Delta t$. The values of these fields at the nodes of the grid $n = 1, \ldots, N$ correspond to the vectors of the $I$-dimensional feature space $\mathbf{f}^{(n)} = \{f_i^{(n)}\}$. A spatio-temporal forecast field $\Phi$ is a function of the fields of features. It is trained using retrospective data: (1) A sample set of target earthquaes $q = 1, \ldots, Q$ with the magnitudes $M \geq M^*$ and (2) a set of grid fields of features $F_i, i = 1, \ldots, I$, which describes spatial (quasi-stationary) and spatio-temporal (dynamic) properties of the seismotectonic process.

The method of the minimum area of alarm uses the following data model.

1. The epicenters of earthquaes with magnitudes $M \geq M^*$ (target events) are preceded by the anomalous (low-probability) values of the fields of features. Let's consider the fields of features to be designed in such a way that for each anomaly, the values of some of these fields are close to their maximum or minimum. To simplify the explanation, we assume that the anomalies refer only to the largest values of the fields of features.

2. If the $f^{(q)}$ is an anomaly vector, preceding the target event $q$, then any vector $f$ with the components $f_i \geq f_i^{(q)}$ for all $i = 1, \ldots, I$ can also precede a similar target event (monotonicity condition).

We will call the base vectors of the feature space the vectors for which $f \geq f^{(q)}$ componentwise. The nodes of the grid of the forecast field $\Phi$ with the values $\phi \geq \phi^{(q)}$ we will call the base nodes of the forecast field.

From the assumption that anomalous refers only to the largest values of the fields of features and the monotonicity condition, it follows that the earthquake forecast can be carried out using the simplest threshold decision rule. If the value of the forecast field $\phi^{(n) \geq \theta}$, then spatio-temporal alarm cylinders are created at all base nodes of the forecast field with the values $\phi \geq \phi^{(q)}$. The alarm cylinder of the grid node $n$ with the coordinates $(x^{(n)}, y^{(n)}, t^{(n)})$ has the center of the base in the node $(x^{(n)}, y^{(n)}, t^{(n)})$, the base radius $R$, and the element $[(x^{(n)}, y^{(n)}, t^{(n)}), (x^{(n)}, y^{(n)}, t^{(n)})]$. From this, it follows that for a given value of the threshold $\theta$ an earthquake with the epicenter coordinates and time $(x^*, y^*, t^*)$ will be detected if and only if the cylinder with the center of the base $(x^*, y^*, t^*)$, the radius $R$, and the element $[(x^*, y^*, t^* - T), (x^*, y^*, t^*)]$ contains at least one grid node with the value $\phi^{(n)} \geq \theta$. This cylinder will be called a precursor cylinder.

The alarm field detects an earthquake if its epicenter falls within an area consisting of a combination of alarm cylinders (alarm area). The quality of the forecast field at threshold $\theta$ is determined by two indicators: (1) The fraction of correctly detected events $Q^*(\theta)$ from all $Q$ events $U(\theta) = Q^*(\theta)/Q$ (probability of detection) and (2) the fraction of number of grid nodes, falling in the alarm area $L^*(\theta)$, from the number of all grid nodes $L$ of the analyzed area $V(\theta) = L^*(\theta)/L$ (alarm volume).

For training, we have a set of target events with magnitudes $M \geq M^*$ and a set of fields. At the first step, the algorithm should move from a set of target earthquaes to a set of target earthquake precursors. A precursor of the earthquake $q$ is the vector $f^{(q)}$ of a feature space which has a minimum

volume of alarm $v^{(q)} = L^{(q)}/L$ among all vectors corresponding to the grid nodes of the precursor cylinder of the event $q$, where $L$ is the number of all grid nodes of the analyzed area, $L^{(q)}$ is the number of nodes in the grid of the alarm area generated by the base points of the vector $f^{(q)}$.

The algorithm for constructing the forecast field is nonparametric. There are the three most important versions of the algorithm. The first version of the algorithm is to construct the forecast field so that when the threshold $\theta$ decreases, the training earthquakes are detected in the sequence in which the corresponding alarm volumes increase $v^{(Q)} \leq v^{(Q-1)} \leq \ldots \leq v^{(2)} \leq v^{(1)}$ (this version is selected for testing). The version consists of the following steps.

1.  To generate a training set $\{f^{(q)}, v^{(q)}\}$, which consists of earthquake precursors $f^{(q)}$ and corresponding alarm volumes $v^{(q)}$.
2.  To sort the precursors $f^{(q)}$, $q = 1, \ldots, Q$, by the alarm volume $v^{(Q)} \leq v^{(Q-1)} \leq \ldots \leq v^{(1)}$ in ascending order.
3.  To assign to the nodes of the grid of the forecast field $\Phi$ a value of 0.
4.  To replace the value of 0 by $Q$ at the nodes of the grid of the forecast field, for which the monotonicity condition, $f_i^{(n)} \geq f_i^{(Q)}$ for all $i = 1, \ldots, I$, is satisfied in the feature space; to replace the value of 0 by $Q - 1$ at the nodes of the grid of the forecast field, for which the monotonicity condition, $f_i^{(n)} \geq f_i^{(Q-1)}$ for all $i = 1, \ldots, I$ is satisfied in feature space, and then, successively, in the same way, to replace the values 0 by $Q + 1 - q$.

Obviously, the choice of the order of the earthquake precursors at the 2nd step of the algorithm determines the dependence $U(V)$ obtained from the forecast field. The 2nd version of the algorithm makes it possible to optimize the forecast field so that when the next target earthquake is detected, the alarm volume increases by a minimum value. To do this, one should arrange the precursors so that, when changing from event detection $q + 1$ to event $q$, the increase in alarm volume is minimal. Here, at each transition from the previously selected event $q + 1$ to $q$, a small search through the remaining $q$ events is required. The 3rd version of the algorithm allows one to optimize the forecast field so that it detects the maximum number of target earthquakes with a total alarm volume of less than or equal to the predetermined value. In this case, you need to perform a full search on the selected number of events. The 3rd version of the algorithm allows optimizing the forecast field so that it detects the maximum number of target earthquakes, provided that the total alarm volume does not exceed the specified value. In this case, you need to perform a full search for the selected number of events.

## 4. Testing

The purpose of testing is to verify the proposed method of the forecast. Testing is carried out in accordance with the known characteristics of the catalog of earthquakes. Exploring the possibility to improve the quality of the forecast using a wider set of characteristics of earthquake catalogs or by adding other sources of input data is beyond the scope of this work.

The method of minimum area of alarm was tested on the platform of automatic earthquake forecast (http://distcomp.ru/geo/prognosis). The system tests the data with a constant step $\Delta t$. On each step (at time $t$) the raster fields of features are computed, the alarm area is trained based on data before the time $t$, and the system tests for time since $t$ till $t + \Delta t$ if the alarm area covers an epicenter of the target earthquake. Then at time $t + \Delta t$, the training time is increased by $\Delta t$, the alarm zone is updated and the test is repeated.

Testing of the forecast method should provide an opportunity to compare different methods of solving the problem on the same indicators of the forecast quality. In this method, we use two quality indicators: The probability of detecting the target events from the test interval $U = Q^*/Q$ and the volume of alarm $V = L^*/L$. The number of target events $Q$ is determined by a set of test samples, the number of target events detected $Q^*$ is determined by the results of the forecast, the analysis area and its size $L$ is selected at the beginning of the test, the size of the alarm zone $L^*$ is determined by the training data.

In the following test experiments, the area of analysis was constructed in the following way: Any point is included in the area if in a circle around this point with radius $R = 100$ km for the period 1984–1993 there are more than 300 earthquake epicenters. This condition allows one to select a seismically active area for analysis but does not ensure its seismic homogeneity. Therefore, the indicator of the volume of alarm obtained during testing should be considered only in the context of the selected area of analysis. At the same time, the choice of the field of analysis according to a formal rule makes it possible to compare the results of the forecast obtained using various methods and according to different data.

One way to assess the quality of a forecast is to compare a regular forecast obtained by the algorithm under analysis (regular forecast) with a random one. We will assume that the forecast is random if the values of the forecast field are selected from a segment in accordance with a uniform distribution. Obviously, for this probabilistic model, the alarm volume $V_r$ is equal to the probability of a random prediction $U_r$. It follows from this that comparing the probability of a regular forecast $U$ with the probability of a random forecast $U_r$ for the same alarm volumes $V = V_r$ is equivalent to comparing $U$ with the corresponding alarm volume $V$. If, at the same time, a sample of target events were cleaned of aftershocks and foreshocks, then by proposing the independence of target events and using the binomial distribution model for them, we could build a confidence interval for estimating $U$.

In the number of articles, the results of a regular forecast are compared with the results of a forecast by a stationary field. In papers [39–41] the regular forecast is compared with the forecast by the 2D field of seismic activity (or earthquake epicenter density). The result of the comparison makes it possible to evaluate the efficiency of a regular forecast in relation to the forecast by the field $F$, which is based only on the spatial heterogeneity of the seismic process. Comparison of results can be done in two ways. In one method the probabilities of regular prediction of $Q$ target earthquakes are compared with the results of predicting the same earthquakes by a stationary field $F$ (for example, $F$ is a 2D field of the earthquake epicenter density). Another method uses the Gutenberg–Richter model [42]. In the beginning, the catalog of earthquakes with the following conditions is constructed: (1) The epicenters of earthquakes are in the area of analysis, (2) the magnitudes exceed the representative, and (3) the depth of the epicenters does not exceed the values specified for the target earthquake. It is assumed that $b$-value is the same for the entire area of analysis and the earthquake catalog agree well with the spatial distribution of seismicity. The alarm field $V(\theta)$ is calculated by the field $F$. Then, in accordance with the alarm field, the dependence $N^*(V)$, is calculated, where $N^*$ is the number of epicenters in the alarm zone, and $V$ is the alarm volume. The dependence $N^*(V)$ is normalized to the number of all $N$ epicenters in the analyzed area. According to the Gutenberg–Richter law and the assumption $b =$const, we have $N^*(V) = C \exp(d^* - bm)$, $N(V) = C \exp(d - bm)$, and $\mu(V) = N^*/N = \exp(d^* - d)$. Thus, the value $\mu(V)$ does not depend on the magnitude of earthquakes. It shows the proportion of earthquakes with a magnitude higher than a given, which fall into the alarm zone. Consequently, the value of $\mu(V)$ in the scope of our model is equal to the probability of forecasting the target earthquakes using the stationary field $F$. If the field $F$ is the density field of the earthquake epicenters, then the field obtained according to the Gutenberg–Richter law is denoted by the letter $\mu$.

Testing was performed for two regions: The Mediterranean and California. The Mediterranean region: $10°–30°$ E, $34°–47°$ N. Input data: Earthquakes for the period from 27.05.1983 till 14.02.2018 with magnitudes $M \geq 2.7$ and depths of hypocenters $H \leq 160$ km from the International Seismological Centre catalog (see Materials and Methods). Target earthquakes: Magnitudes $M \geq 6.0$ and hypocenter depths $H \leq 60$ km. California region: $126°–114°$ W, $32°–43°$ N. Input data: Earthquakes for the period of 01.01.1983–15.02.2018 with magnitudes $M \geq 2.0$ and depths of hypocenters $H \leq 160$ km from the NEIC USGS catalog (see Materials and Methods). For the forecast, the target earthquakes with magnitudes $M \geq 5.7$ have been selected.

The following six fields of features were analyzed for forecasting:

- $F_1$ is the 3D field of the density of all considering earthquakes in the region.
- $F_2$ is the 3D field of mean magnitudes among all considering earthquakes in the region.

The estimation of 3D fields of $F_1$ and $F_2$ is performed with the method of local kernel regression. The kernel function for the *n*-th earthquake has the form $K_n = [\cosh^2(r_n/R)^2 \cosh^2(t_n/T)]^{-1}$, where $r_n < R\epsilon$, $t_n < T\epsilon$ are the distance and time interval between the *n*-th epicenter of the earthquake and the node of the 3D grid of the field, $\epsilon = 2$, $R = 50$ km, $T = 100$ days for $F_1$ and $R = 100$ km, $T = 730$ days for $F_2$.

- $F_3$ is the 3D field of negative temporal anomalies of the density of earthquakes.
- $F_4$ is the 3D field of positive temporal anomalies of the density of earthquakes.
- $F_5$ is the 3D field of positive temporal anomalies of mean earthquake magnitude.

To estimate the field of $F_3$, $F_4$, $F_5$, the Student's *t*-statistic was used, which is defined for each grid node as the ratio of the difference of average values of the current (196 days) and background (3650 days) intervals to the standard deviation of this difference. Positive t-statistics values correspond to higher values on the test interval.

- $F_6$ is the 2D field of the density of earthquake epicenters: Kernel smoothing in the interval 1988–2008 the parameter $R = 50$ km.

The grid fields for the Mediterranean were calculated in a grid step $\Delta x \times \Delta y \times \Delta t = 0.2° \times 0.13° \times 49$ days. The forecast field was trained from 1998 until the next step of the forecast after 2008. The radius of the alarm cylinder is $R = 20$ km, and the element is $T = 50$ days. Testing is performed in 2008–2019. There are 11 target earthquakes in the analysis area. We used the method of stepwise selection to find the most informative fields of features. The algorithm selected the $F_3$ and $F_6$ fields to construct the alarm field.

We compare the earthquake prediction probabilities obtained using different fields of features in Table 1: $U_1$ is the forecast probability using the earthquake density field 2D ($F_6$), $U_2$ is the probability using the 3D field of negative earthquake density anomalies ($F_3$), $\mu(V)$ is the probability of forecast by 2D field of earthquake epicenters density, obtained using the Gutenberg–Richter model, and $U_3$ is the probability using $F_3$ and $F_6$ fields. We can see that the highest probability of a successful forecast occurs when the fields $F_3$ and $F_6$ are used together. When $V = 0.2$ ($U_r = 0.2$), the ratios for the prediction probability obtained with $F_3$ and $F_6$ fields to the prediction probabilities obtained with 2D earthquake density field ($F_6$) and for the field calculated using the Gutenberg–Richter ratio, are equal respectively $U_3(0.2)/U_1(0.2) = 0.91/0.64 = 1.49$ and $U_3(0.2)/\mu(0.2) = 0.91/0.41 = 2.2$. Table 1 shows the values of two types of alarm volumes: $V_{learn}$ is the alarm volume received in accordance with the training data, and $V_{test}$ is the alarm volume corresponding to the alarm volume $V_{learn}$ but observed on the test data. You can see that when testing in almost all cases, except for testing the 2D-field $F_6$, the volumes of $V_{test}$ are greater than $V_{learn}$. This is explained by the fact that the number of recorded earthquakes in a region changes over time (Figure 1). The number of earthquakes is influenced by the development of a seismic network and natural changes in the seismic process. The Figure 1 shows that the number of earthquakes increases significantly in the test interval. The same anomaly appears on the plot of the time series of average values of the density of earthquake epicenters throughout the analysis area (Figure 2). An increase in the density of earthquake epicenters leads to an increase in the field values of the $F_3$ function, which ultimately leads to an increase in the volume of anxiety during testing.

The grid fields for California were calculated in a grid step $\Delta x \times \Delta y \times \Delta t = 0.125° \times 0.11° \times 49$ days. The radius of the alarm cylinder is $R = 14$ km, and the element is $T = 100$ days. Testing of the earthquake forecast was performed for the interval 2009–2018. There were nine target earthquakes. The algorithm selected tree fields of features for the construction of the alarm field: $F_4$, $F_5$, and $F_6$.

Table 2 shows the probabilities of earthquake forecast for California.

Figures 3 and 4 show the test results for both regions. They depicted polygons selected as the area of analysis, and circles are the target epicenters of earthquakes in 2009–2018 with $M \geq 6.0$ for Mediterranean and $M \geq 5.7$ for California.

**Table 1.** Comparison of the probabilities of earthquake forecast for the Mediterranean.

| $V_{learn}$: Alarm Volumes for Learning Interval | 0.01 | | 0.05 | | 0.1 | | 0.15 | | 0.2 | |
|---|---|---|---|---|---|---|---|---|---|---|
| Test indicators | $V_{test}$ | $U_{test}$ | $V_{test}$ | $U_{test}$ | $V_{test}$ | $U_{test}$ | $V_{test}$ | $U_{test}$ | $V_{test}$ | $U_{test}$ |
| Field $F_6$ | 0.00 | 0.00 | 0.03 | 0.09 | 0.10 | 0.09 | 0.15 | 0.45 | 0.20 | 0.64 |
| Field $F_3$ | 0.00 | 0.00 | 0.02 | 0.09 | 0.20 | 0.36 | 0.29 | 0.45 | 0.35 | 0.64 |
| $\mu(V)$: probability for the field $F_6$ obtained by the model | - | 0.00 | - | 0.1 | - | 0.26 | - | 0.35 | - | 0.41 |
| Fields $F_3$ and $F_6$ | 0.03 | 0.00 | 0.08 | 0.36 | 0.15 | 0.45 | 0.24 | 0.55 | 0.32 | 0.91 |

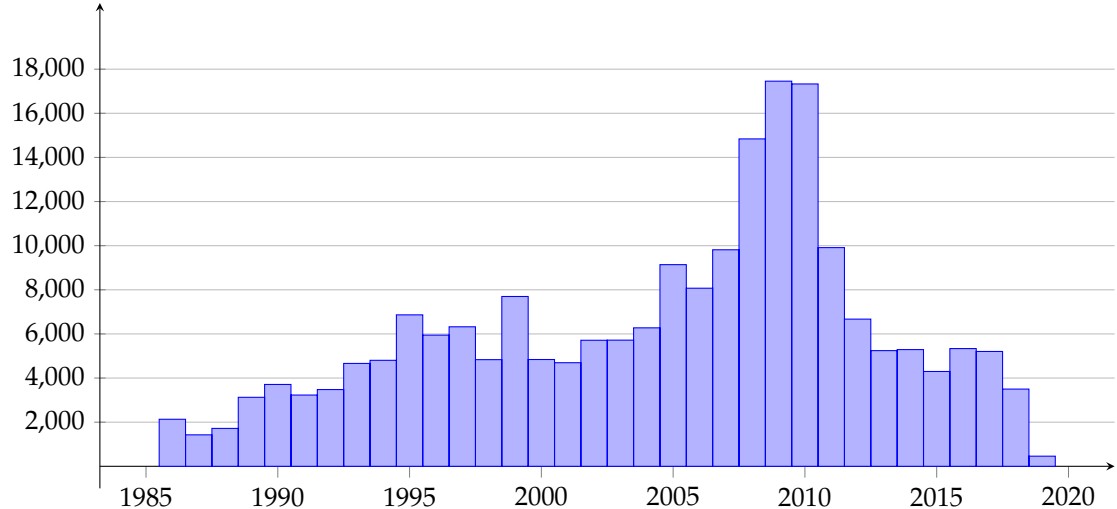

**Figure 1.** Histogram of the number of earthquakes with a magnitude greater than 2.7 and a depth of hypocenters less than 160 km.

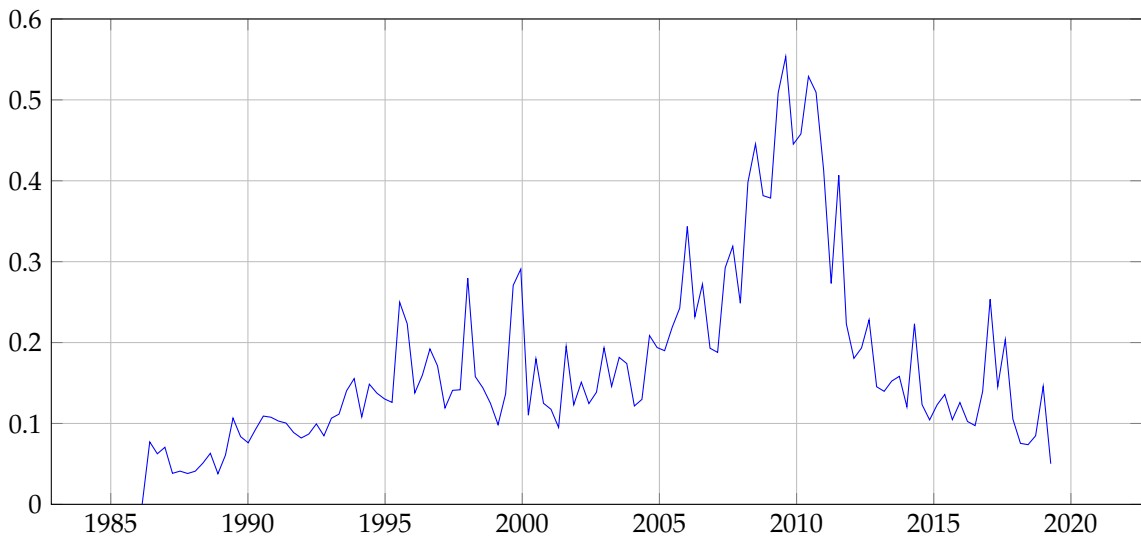

**Figure 2.** Time series of average values of the density of earthquake epicenters throughout the analysis area (trend line).

**Table 2.** Probabilities of earthquake forecast for California.

| Volume of Alarm $V_{learn}$ | Volume of Alarm $V_{test}$ | Number of Correct Forecasts | Forecast Probability $U$ |
|---|---|---|---|
| 0.01 | 0.01 | 1 | 0.11 |
| 0.05 | 0.06 | 4 | 0.44 |
| 0.1 | 0.13 | 4 | 0.44 |
| 0.15 | 0.13 | 4 | 0.44 |
| 0.2 | 0.25 | 8 | 0.89 |

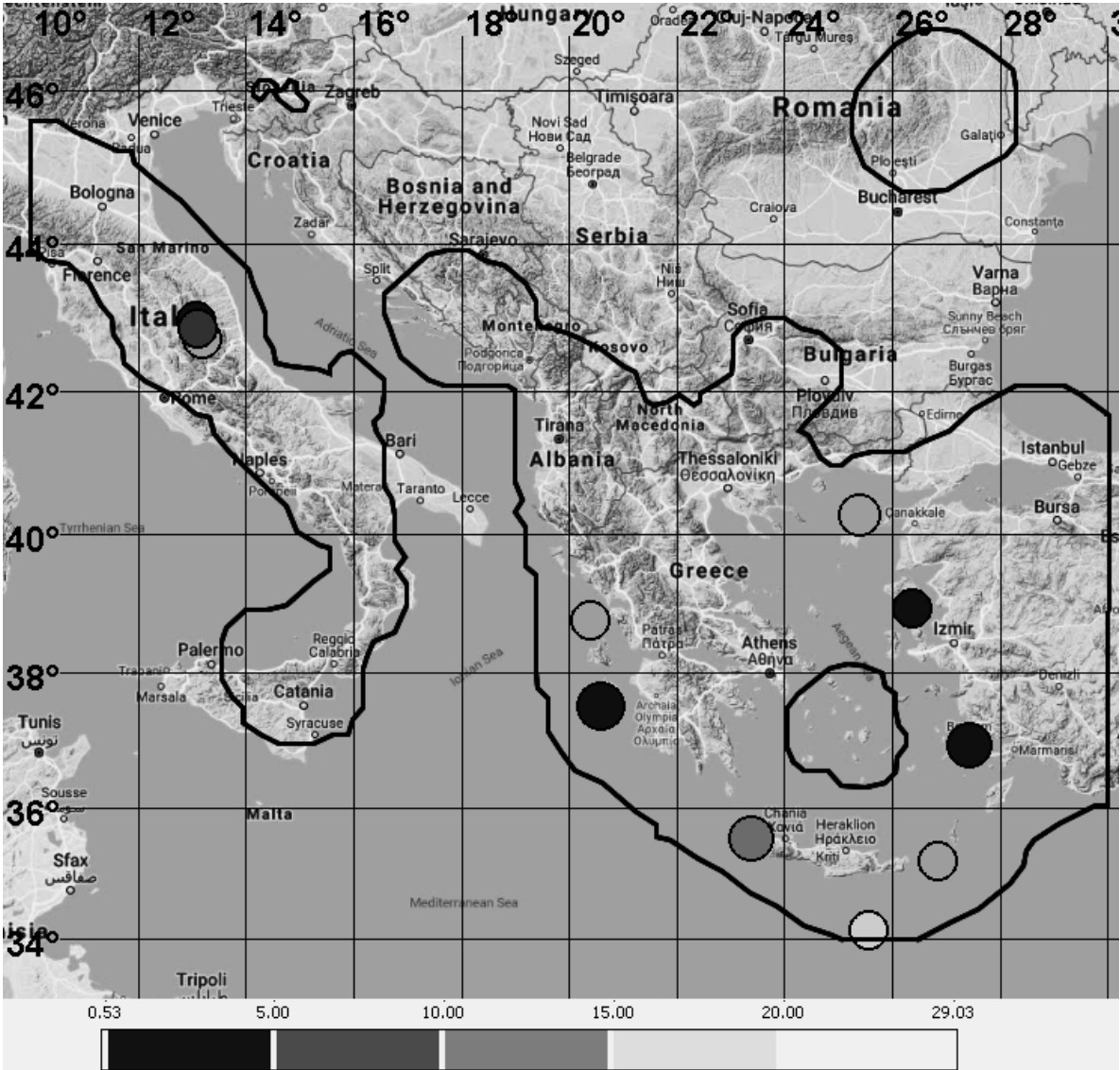

**Figure 3.** Area of analysis (marked with thick black line) and tested target epicenters of earthquakes in 2009–2018 in the Mediterranean region. Shades of grey indicates the minimum volume of alarm with which the epicenter was forecasted. Darkness of grey decreases in accordance with the volume of the alarm: 0.05, 0.1, 0.15, 0.2. A white color indicates that an earthquake is not forecasted with an alarm volume of less than 0.2.

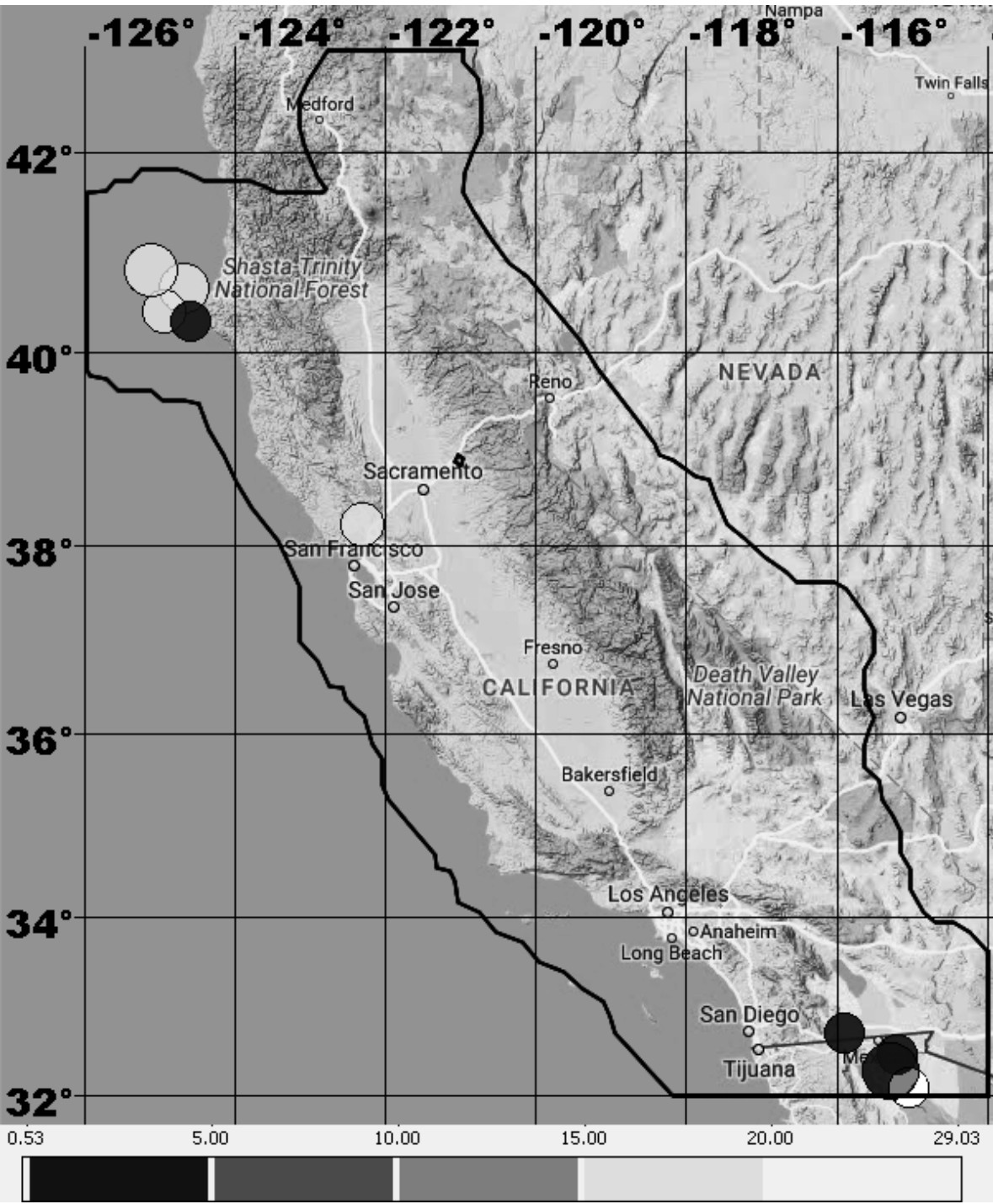

**Figure 4.** Area of analysis (marked with thick black line) and tested target epicenters of earthquakes in 2009–2018 in California. Shades of grey indicates the minimum volume of alarm with which the epicenter was forecasted. Darkness of grey decreases in accordance with the volume of the alarm: 0.05, 0.1, 0.15, 0.2. A white color indicates that an earthquake is not forecasted with an alarm volume of less than 0.2.

## 5. Discussion

The method of approximating the interval expert estimates compiles the $M_{max}$ map, assuming a repetition of strong earthquakes and the existence of a relationship between $M_{max}$ and the properties of the geological environment $x = (x_1, \ldots, x_I)$. At first, experts independently estimate $M_{max}$ values in a set of the most studied points of a region. The algorithm approximates the dependence of $M_{max}$ on

geological and geophysical features by the function $F(\mathbf{x})$. The dependence $F(\mathbf{x})$ is defined as the sum of the non-linear functions of each of the feature. The $M_{max}$ map is the $F(\mathbf{x})$ values calculated for the whole region. The presence of the formal forecast rule $F(\mathbf{x})$ allows the expert to study the contribution to the forecast of each feature and interpret the map as the sum of the nonlinearly transformed feature fields.

The method of the minimum area of alarm solves the problem of one-class classification. The method algorithm has two peculiar properties. The first relates to the data model. The model postulates two properties of anomalous objects: (1) Anomalous objects are unlikely, and some of their properties take values close to the maximum (or minimum) among the sample, and (2) the vectors of the space of features, which are componentwise larger (or smaller) of the vector corresponding to the anomalous object, can also be anomalous objects. Both these properties seem sufficiently natural. This model allows one to build a classification rule from a set of anomalous objects. In this case, normal objects are taken into account statistically through the probability of detecting anomalous objects by a random forecast. The second difference is that the algorithm allows constructing a forecast function that optimizes the probability of detecting anomalous objects in the training sample if the probability of a random forecast is not more than the predetermined value.

## 6. Conclusions

We considered two machine learning methods and their implementations to seismic hazard forecast. The method of approximation of interval expert estimates of $M_{max}$ demonstrated good seismic zoning for many seismically active regions. The method of the minimum area of alarm is the basis of an automatic earthquake forecast system. The considered results of testing suggest that the method and the forecast system might contribute to advance in the problems of earthquake forecasting.

## 7. Supplement

The method of minimum area of alarm is the basis of an automated web-based platform that systematically forecasts target earthquakes. We presented the results of testing the approach to earthquake prediction in the Mediterranean and Californian regions. The goal of the test was to analyze the approach, the machine learning method, and the earthquake prediction platform. For the tests, ordinary parameters of earthquake catalogs were used. The testing was performed on data that did not participate in the training and showed a satisfactory forecast quality for both regions. The web-based platform has been launched and automatically calculates the seismic hazard fields from February 2018 [43]. During the time from 1 February 2018 to 8 July 2019, four target earthquakes occurred in these regions. In the Mediterranean region, two epicenters were predicted and fell into an area with an alarm volume of up to 15% (Figure 5), and in the California region, two epicenters fell into an area with an alarm volume greater than 20% and were not predicted (Figure 6).

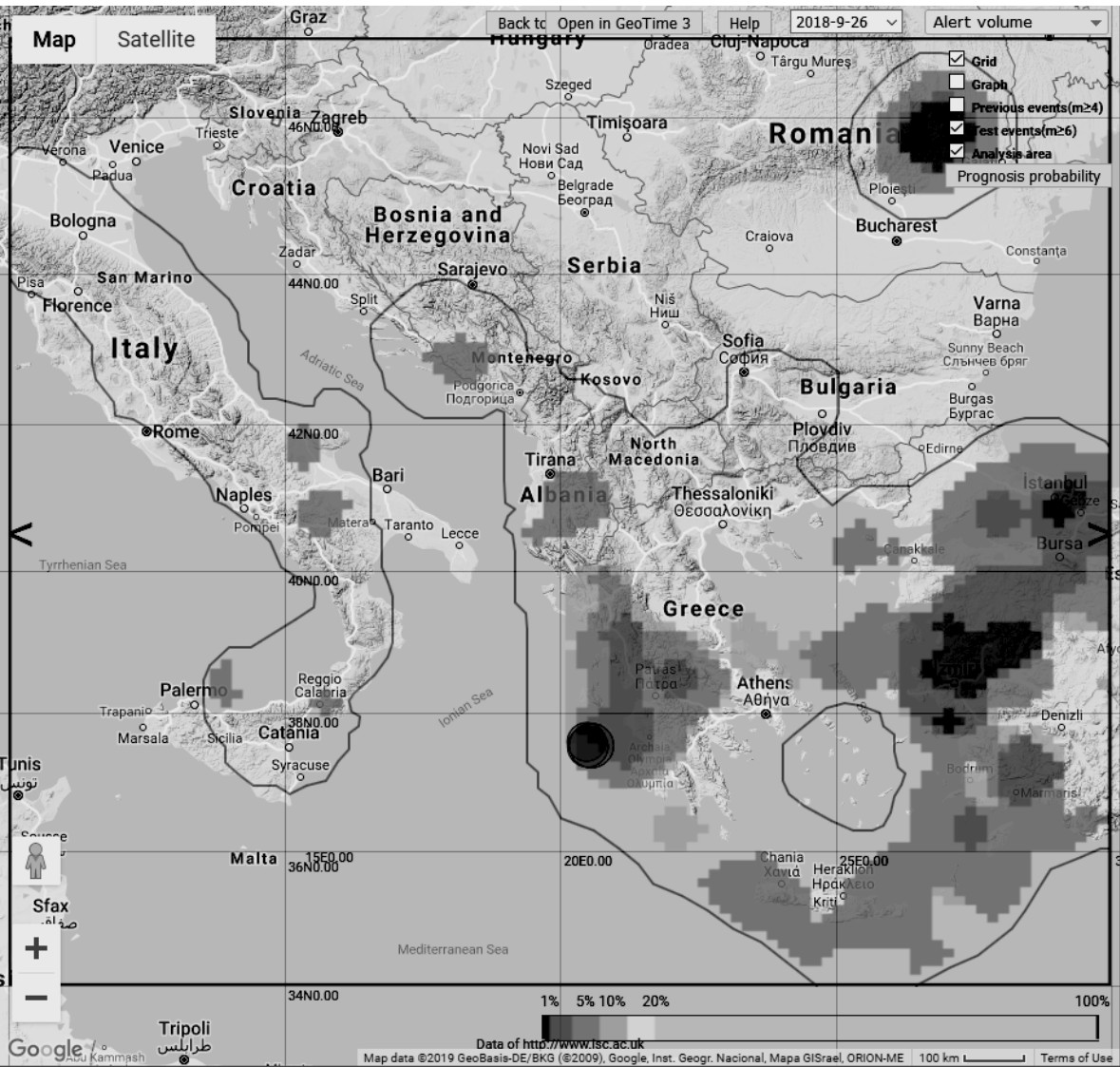

**Figure 5.** Screenshot of the web-based platform working window in the Mediterranean region: The map shows the alarm zone for earthquakes with a magnitude $M \geq 6.0$ and the predicted epicenters of earthquakes of 25 October 2018 with a magnitude of 6.6 and 30 October 2018 with a magnitude of 6.2 calculated for training according to data up to 26 September 2017. The palette shows areas with different alarm volumes in percent: 1%, 5%, 10%, 15%, 20%.

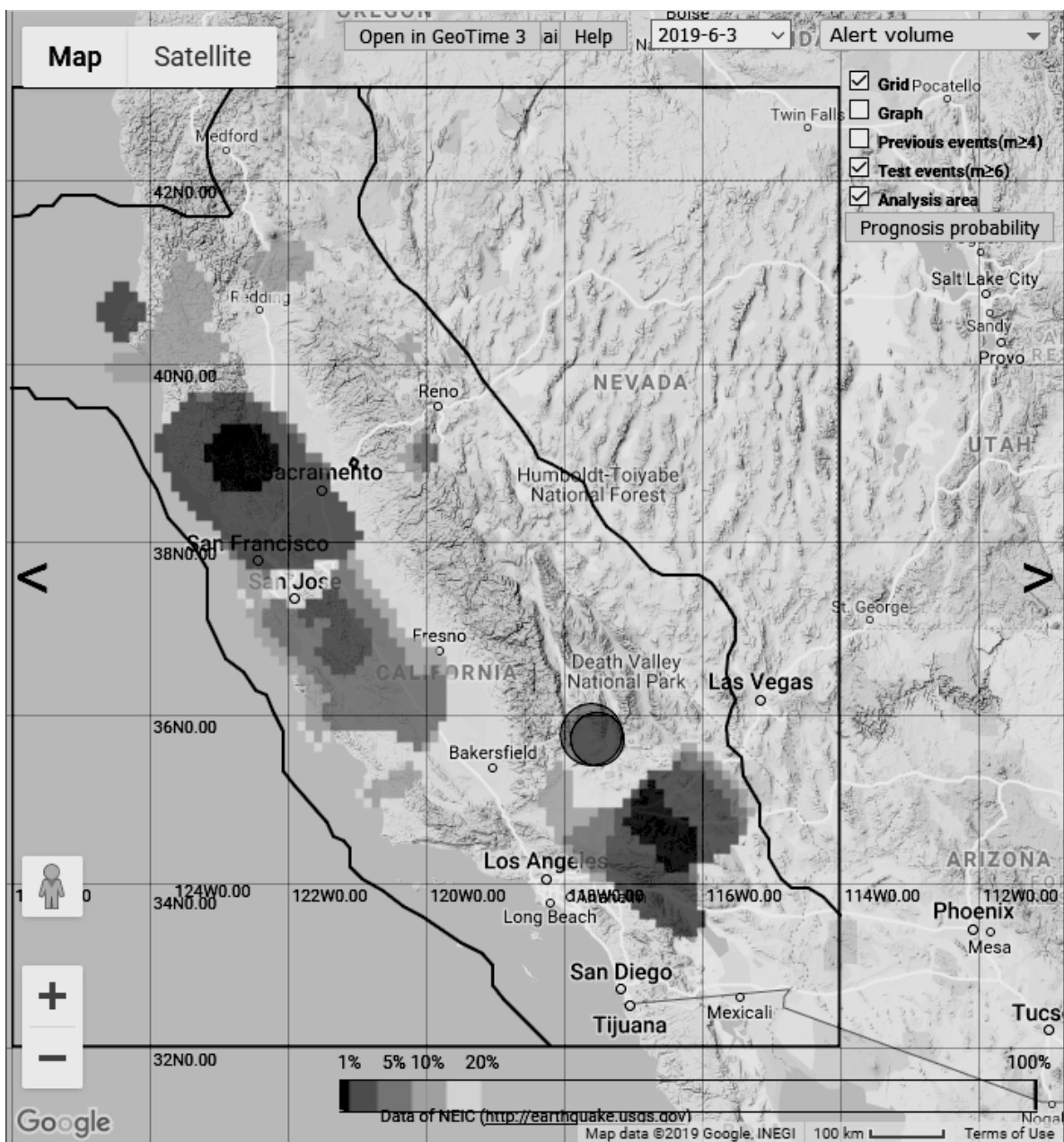

**Figure 6.** Screenshot of the web-based platform working window: The map shows the alarm zone for earthquakes with a magnitude $M \geq 5.7$ and the epicenters of earthquakes of 4 July 2019 with a magnitude of 6.4 and 6 July 2019 with a magnitude of 7.1 calculated during training according to data until 3 June 2019. The palette shows areas with different alarm volumes in percent: 1%, 5%, 10%, 15%, 20%.

## 8. Materials and Methods

International Seismological Centre [44] was searched using http://www.isc.ac.uk/iscbulleti n/search/bulletin/ (last accessed on 14 February 2018). This online catalog was selected for its robustness and universality. It combines data from a lot of catalogs, and every earthquake with a magnitude more than three is manually checked. NEIC USGS catalog [45] was searched using https://earthquake.usgs.gov/earthquakes/feed/ (last accessed on 15 February 2018). This catalog was chosen because of numerous registered small earthquakes (magnitude of completeness less than 1.5) in California. Plots were made using the GeoTime 3 (www.geo.iitp.ru/GT3; [8]).

**Author Contributions:** Conceptualization, V.G.G.; Formal analysis, V.G.G. and A.B.D.; Investigation, V.G.G.; Methodology, A.B.D.; Project administration, V.G.G.; Visualization, A.B.D.; Software, A.B.D.; Writing—original draft, V.G.G.; Writing—review & editing, V.G.G. and A.B.D.

**Funding:** The paper is supported by the Russian Foundation for Basic Research, project 17-07-00494.

**Conflicts of Interest:** The authors declare no conflict of interest.

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
