# Peer review of "Machine Learning Methods for Seismic Hazards Forecast"

_geosciences, doi:10.3390/geosciences9070308_

Round 1

Reviewer 1 Report

The paper deals with the development of machine learning techniques, which are innovative, and in connection with future hazard in earthquake prone areas, and for this reason is important. There are, however, several points inside the paper that need additional work and corrections. Specific comments are reported, which I hope will contribute to the improvement of its revised version.

MAJOR COMMENTS

1.                  A careful reading of the text will result to the improvement of syntax and grammar. In many places, slight modifications are necessary for the text to be conceivable.

2.                  Line 24: Do you need geological properties? Even in seismically inactive areas? On the other hand, the degree of active deformation and faulting type don’t play any role?

3.                  Lines 240 – 243: These data samples are not complete. Please support your selection and present arguments.

SPECIFIC COMMENTS

1.     Line 2: Better “whereas” instead of “while”

2.     Line 8: Better “identify” instead of “learn”

3.     Line 11: “… were used” – better after the subject, at the end of the sentence.

4.     Line 20: “… aerospace observations” – Please, be more specific and give reference

5.     Line 21: “… completeness of the description” …? Please, clarify!

6.     Line 24: “… aims to build seismic zoning maps” – Either clarify, or change to that seismic zoning is prerequisite for seismic hazard assessment.

7.     Line 26: “… cannot be measured instrumentally” – Why not? Please, clarify!

8.     Line 40: … “good” estimates: could you find a more proper expression?

9.     Lines 59 – 60: “The task of predicting an earthquake is to determine the its time, location and magnitude of a future earthquake” – suggested corrections

10.  Line 61: “… mathematical models” – better “stochastic models”

11.  Line 83: delete the duplicated “the”

12.  Line 86: “… less than or greater than …” – suggested correction

13.  Line 114: “… process are be described …” – suggested correction

14.  Line 118: “… data that contains …” – suggested correction

15.  Lines 123 – 125: Please, rewrite this sentence – it is not clear.

16.  Line 139: “… epicenter coordinates” – not only epicenters, time is also included

17.  Line 154: “… are three the most …” – better “… are the three most …”

18.  Lines 200 – 201: Please, rephrase!

19.  Lines 224 – 226: Please, make this statement more clear. Perhaps with more sentences.

Author Response

All answers are in word file

Reviewer 2 Report

The Authors  address an interesting approach for seismic hazards forecast: however, some major revision should be considered in order to make this work more appreciated by the reader.

Introduction, Chapter 2 and Chapter 3:

The two machine learning methodologies should be described  more extensively and correlated with some figures and flowcharts which would help the reader to better appreciate the proposed approaches.

Several formulas do not have a complete description of all the variables.

Sentences like ‘in the opinion of the qth expert’ should be better elaborated, in order to permit the reader to fully appreciate the methodologies.

A detailed analysis of the robustness of the obtained results is missing: the quantification of the error estimates associated to each machine learning methodology should be included in the text.

Chapter 4 (Testing)

The quality of existing Tables and figures should be improved. The whole text should be better structured and possibly complemented with additional tables and figures to help the reader to focus on the main results.

Discussion

The limits and potentials of the methodologies should be discussed in more detail. The Authors should reconsider the sentence ‘the forecast system can be used to solve the problems of earthquake forecasting’: earthquakes’ prediction is a very difficult challenge, which cannot be solved so easily. The proposed methodologies might contribute to advance in this challenge: to solve it is very ambitious.

I would like to encourage the Authors to revise their work in order to help the reader to fully appreciate it.

Author Response

All answers are in word file

Round 2

Reviewer 2 Report

The Authors have incorporated some sentences helping the reader to appreciate more the overall approach. The revision of the Discussion lead to a more realistic final sentence.

There are just a few minor spelling typos to be checked.

I still believe that some figures and tables could be improved to help the Authors to better present their study: yet, I will leave the final decision on this point to the Editors.

I would like to thank the Authors for their efforts and encourage them to continue to further develop their studies.

Author Response

We carefully checked the paper and make a lot of small changes (mainly insert omitted articles). Also, we redraw images to be more transparent for the reader and submitted two of them as vector images.

Thank you for your revision!